# De Novo Genome Assembly at Chromosome-Scale of *Hermetia illucens* (Diptera Stratiomyidae) via PacBio and Omni-C Proximity Ligation Technology

**DOI:** 10.3390/insects15020133

**Published:** 2024-02-17

**Authors:** Simone Costagli, Linda Abenaim, Giulia Rosini, Barbara Conti, Roberto Giovannoni

**Affiliations:** 1Department of Biology, University of Pisa, Via Derna 1, 56126 Pisa, Italy; s.costagli3@studenti.unipi.it (S.C.); giulia.rosini@biologia.unipi.it (G.R.); roberto.giovannoni@unipi.it (R.G.); 2Department of Agriculture, Food and Environment, University of Pisa, Via del Borghetto 80, 56124 Pisa, Italy; linda.abenaim@phd.unipi.it; 3Nutrafood Center, University of Pisa, Via del Borghetto 80, 56126 Pisa, Italy; 4CIRSEC, Center for Climate Change Impact, Centro di Ricerche Agro-Ambientali, University of Pisa, 56126 Pisa, Italy

**Keywords:** *Hermetia illucens*, de novo genome assembly, topology associated domains, gene annotation, *Larval serum protein-2*

## Abstract

**Simple Summary:**

A new chromosome-scale genome assembly for *Hermetia illucens* was achieved by combining PacBio and Omni-C proximity ligation technologies. The final genome size was 888.59 Mb, with a scaffold N50 value of 162.19 Mb and a BUSCO completeness of 89.1%. By utilizing Omni-C proximity ligation technology, key topological features such as topologically associated domains influencing gene expression regulation were identified. About 65.62% of genomic sequences were identified as repeated sequences, and the MAKER pipeline annotated 32,516 genes. The annotated *H. illucens Lsp-2* genes were further characterized, and the three-dimensional organization of the encoded proteins was predicted. This new high-quality chromosome-scale genome assembly facilitates further genotypic and phenotypic characterization of *H. illucens*, a species with significant relevance to various biotechnological applications.

**Abstract:**

*Hermetia illucens* is a species of great interest for numerous industrial applications. A high-quality reference genome is already available for *H. illucens*. However, the worldwide maintenance of numerous captive populations of *H. illucens*, each with its own genotypic and phenotypic characteristics, made it of interest to perform a de novo genome assembly on one population of *H. illucens* to define a chromosome-scale genome assembly. By combining the PacBio and the Omni-C proximity ligation technologies, a new *H. illucens* chromosome-scale genome of 888.59 Mb, with a scaffold N50 value of 162.19 Mb, was assembled. The final chromosome-scale assembly obtained a BUSCO completeness of 89.1%. By exploiting the Omni-C proximity ligation technology, topologically associated domains and other topological features that play a key role in the regulation of gene expression were identified. Further, 65.62% of genomic sequences were masked as repeated sequences, and 32,516 genes were annotated using the MAKER pipeline. The *H. illucens* Lsp-2 genes that were annotated were further characterized, and the three-dimensional organization of the encoded proteins was predicted. A new chromosome-scale genome assembly of good quality for *H. illucens* was assembled, and the genomic annotation phase was initiated. The availability of this new chromosome-scale genome assembly enables the further characterization, both genotypically and phenotypically, of a species of interest for several biotechnological applications.

## 1. Introduction

*Hermetia illucens* (Linnaeus, 1758) (Diptera Stratiomyidae), also known as Black Soldier Fly (BSF), is an insect native to the Neotropical region but, due to its widespread presence throughout the world, currently considered a cosmopolitan species [1]. The adults (Figure 1 left) do not feed and can live owing to the larval reserves; they do not bite or sting and are not vectors of pathogens [2]. The larvae (Figure 1 right) are efficient bio-converters of organic substrates and can digest more than double their weight per day [3]. Owing to its high reproduction rate, adaptability to different environmental conditions, and larval voracity, it is currently a much-studied species, especially for its massive larval and pupal production, for industrial application [4]. In particular, the BSF larvae that develop in agricultural organic wastes but also on animal and plant residues can produce frass that can be used as fertilizer, comparable in quality and chemical composition to the commercial ones [5].

Besides its bioconversion, *H. illucens* larvae might also be exploited to obtain derivatives with a high application value, such as protein, lipids, peptides, amino acids, chitin, and vitamins. The *H. illucens* larvae are rich in lipids (about 15–49% of their weight) [6] and are studied to produce biodiesel as an alternative renewable energy source [7]. The BSF biodiesel complies with the European Regulation on the characteristics of biodiesel EN14214, and it has more oxidative stability than rapeseed oil owing to its high content of methyl esters of fatty acids [8]. Furthermore, the BSF pupae, owing to its high chitin content, which represents 8–9% of the biomass, could be used as an alternative source of crustacean chitin [9]. Chitin and its main derivative, chitosan, have important applications in various fields (including biomedical, pharmaceutical, agricultural, textile, and food industries) [10]. Moreover, *H. illucens* larvae, when reared on wastes, are usually exposed to an abundance of various microorganisms, and produce antimicrobial peptides (AMPs) for protection against harmful bacteria [11,12]. Currently, about 57 *H. illucens* peptides with antimicrobial, antitumoral, and antiviral activities have been identified, belonging principally to the cecropin and defensin family [13], and some of these are identified as being responsible for a notable reduction of *Escherichia coli* and *Salmonella* sp. in manure on which the BSF larvae were reared [14,15].

However, the most promising use of *H. illucens* is as animal feed. In fact, owing to its protein and lipid contents (37–63% and 7–39% in larvae dry matter, respectively), BSF meal and oil are considered an animal-grade alternative to fish meal and fish oil usually used to feed fish, chickens, and other animals [16,17]. The EU Regulation 2017/893 indicates that *H. illucens* is one of the seven insect species currently reared in the European Union for feed production.

Owing to its high nutritional content, its ability to recycle organic waste, and its ability to produce derivatives of great importance, *H. illucens* is one of the most exploited species of insects, even though from a genetic point of view, there are still many obscure points to clarify, for which reason the genetic research on the BSF needs to be thorough. Currently, for *H. illucens*, three genome assemblies are available on NCBI (Assembly = GCA_905115235.1, GCA_009835165.1, GCA_001014895.1). Among these, the most recent version now recognized as a reference genome is the assembly GCA_905115235.1. The reference genome is the first chromosome-scale assembly available for *H. illucens*, and it was obtained by using Pacific Bioscience, 10x Genomics linked read, and high-throughput chromosome conformation capture sequencing technology (Hi-C) [18]. Although the current reference genome is a high-quality genome assembly, it was of interest to perform a de novo genome assembly for *H. illucens* by considering the current availability of technologies that can overcome some of the limitations of Hi-C sequencing, and especially as there are no inbred lines available for *H. illucens*. Hi-C technologies provide long-range information about the grouping and linear organization of sequences along entire chromosomes and so can be used for assigning, ordering, and orienting genomic sequences in a de novo genome assembly [19]. However, restriction enzyme (RE)-based Hi-C methods do not digest chromatin regions with a low density of RE, and this does not make the genomic coverage of the libraries obtained too high, it reduces the efficiency of each sequencing run, and it makes the scaffolding of contigs during genome assembly dependent on the density of the RE site. Dovetail Genomics has developed a new proximity ligation protocol called Omni-C. Omni-C is an endonuclease-based protocol for generating Hi-C libraries in a sequence-independent manner, reducing biases imposed by RE site density and making the coverage of the generated libraries more uniform across the genome. («Omni-C^®^ Assay Technical Note». https://dovetailgenomics.com/wp-content/uploads/2021/09/Omni-C-Tech-Note.pdf, accessed on 20 July 2022). The absence of inbred lines for *H. illucens* means that numerous captive populations of *H. illucens* are maintained worldwide, each associated with different genotypic and phenotypic characteristics. For these reasons, performing a de novo genome assembly on the population reared at the Department of Agriculture, Food and Environmental (DAFE) of the University of Pisa was of interest as it will allow us in the future to better characterize this population of *H. illucens* from genotypic and phenotypic points of view.

As described above, the use of *H. illucens* as animal feed is very promising due to its excellent nutritional properties for various livestock diets. However, for this insect species, prospects of genome editing aimed at the insertion of exogenous gene sequences are highly interesting, either to further improve its nutritional characteristics or to make it a vehicle for other proteins not normally expressed by *H. illucens* but associated with positive and beneficial, when not even protective or anti-inflammatory, properties [20]. To ensure high-level expression in tissues and life-cycle stages of greater interest for animal feed use, while minimizing the impact that the expression of an exogenous protein could have on the insect’s normal physiology, it is desirable that these genome editing strategies be site-specific to ensure extremely controlled temporal expression. Specifically, considering that the life cycle stages of *H. illucens* most frequently proposed for use as animal feed are the prepupa and mature larvae, it would be important to ensure the expression of the exogenous genes in these life-cycle stages through site-specific genome editing at gene sequences characterized by this expression profile. After searching for genes with this expression profile in *D. melanogaster* on the FlyBase database (http://www.flybase.org, 10.1093/genetics/iyac035, accessed on 10 June 2022), among the genes of interest, *Larval serum protein 2* (*dmeLsp-2*, Gene ID: 45326) was selected. *DmeLsp-2* has a gene expression profile coherent with that described above, and it encodes a single protein isoform, which can be an advantage, considering that gene coding for several isoforms can more easily be associated with unknown expression regulatory systems that can cause a different expression from the desired one. For this reason, in the *H. illucens* de novo genome assembly reported in this work, those genes with significant similarity to *dmeLsp-2* were more thoroughly investigated. Little is known about the functions of *dmeLsp-2* protein: it has been reported that *dmeLsp-2* is involved in motor neuron axon guidance, although it has also been suggested that LSPs serve as general storage proteins (e.g., as a store of amino acids or energy during metamorphosis) and that they may be incorporated into the cuticle and involved in melanization [21,22,23,24]. In *H. illucens*, *Lsp-2* (*hiLsp-2*) has not yet been characterized, and no bibliographic data are available.

Here, we present a chromosome-scale, 0.888 Gb genome assembly for *H. illucens*, obtained by combining Pacific Bioscience (PacBio) long reads with Dovetail Genomics Omni-C technology. Starting from the chromosome-scale *H. illucens* genome assembled by us, we manually annotated a gene with similarity to *dmeLsp-2*. These resources will be important to improve the characterization of *H. illucens* and increase its potential biotechnological applications.

## 2. Materials and Methods

### 2.1. Hermetia Illucens Mass Rearing

*Hermetia illucens* was reared under laboratory conditions (T. 30 °C, R.H. of 60%, photoperiod 16:8 = L:D, and radiance of 1500 Lux) from November 2018 at DAFE. The adults were placed in cages (47.5 × 47.5 × 93 cm; model, BugDorm-4M4590DH (Specimen Handling Cage, Mega View Science Co., Ltd., Taichung, Taiwan) in polyester and with a knitted mesh for the exchange of air. An LED panel lamp of 1500 Lux was located over each cage. Each cage of adults was provided with water and sugar ad libitum, and with a plant branch as support to increase the surface available for adults to lean on and perform their “lekking behavior” for mating. Moreover, wooden support for oviposition (20 × 3 × 1 cm) placed on a box containing organic material in decomposition (usually apples) was provided as well in the adult cages. The eggs laid into the groove of the wooden support were removed after 2 days from oviposition with a wet brush with a fine tip and moved into the egg’s hatching area. This area was made of plastic boxes (29 × 18 × 9 cm) containing an artificial diet for larval feeding composed of chicken feed and water (40:60, respectively). After egg hatching, to avoid overpopulation and, therefore, a possible shortage of food, about 1000 six-day-old larvae were moved into the larval rearing area in different containers containing the same feed substrate. Each plastic box was covered with gauze to allow for air exchange and to ensure humidity. Once the larvae reached the pupal stage, the gauze was removed, and the uncovered plastic boxes were positioned inside the rearing cages for the emergence of the new generation of adults.

### 2.2. Genome Library Construction and Sequencing

All de novo genome sequencing was carried out by Dovetail Genomics. High-molecular-weight DNA was extracted from 3 *H. illucens* degutted males, and a PacBio circular consensus sequencing (CCS) library was produced and sequenced on the PacBio™ HiFi™ platform (PacBio, San Diego, CA, USA). A Dovetail Omni-C library was also prepared. Chromatin was fixed in place with formaldehyde in the nuclei of an *H. illucens* adult male and then extracted. Fixed chromatin was digested with DNAse I, and chromatin ends were repaired and ligated to a biotinylated bridge adapter followed by proximity ligation of adapter-containing ends. After proximity ligation, crosslinks were reversed, and the DNA was purified. The purified DNA was treated to remove biotin that was not internal to ligated fragments. Sequencing libraries were generated using NEBNext Ultra enzymes and Illumina-compatible adapters (New England Biolabs, Ipswich, MA, USA). Biotin-containing fragments were isolated using streptavidin beads before the PCR enrichment of each library. The Dovetail Omni-C library was then sequenced on an Illumina HiSeqX platform (Illumina, San Diego, CA, USA).

### 2.3. Genome Assembly

Sequencing data of the PacBio CCS library were used as an input on Hifiasm (v0.15.4-r347), a de novo assembler specifically developed for HiFi reads that can faithfully represent the haplotype information in the assembly and produce good assemblies for diploid genomes [25]. The initial draft assembly was aligned with blast (v2.9.0) against the nt database, blast results were used as input for blobtools (v1.1.1), and scaffolds identified as possible contamination were removed from the assembly [26]. The blobtools-filtered draft assembly was purged of haplotigs and contig overlaps with purge_dups (v1.2.5) [27] to produce the primary filtered assembly. The primary filtered assembly and Omni-C reads with MQ > 50 were used as input data for HiRise (version 2.0.5), a software pipeline designed specifically for using proximity ligation data to scaffold genome assembly with long-range accuracy and contiguity [28], to produce the final assembly.

### 2.4. Genome and Gene Prediction Quality Evaluation

The quality of assemblies returned by Hifiasm and HiRise pipelines was determined by assessing genome completeness with BUSCO (Benchmarking Universal Single-Copy Orthologs, v5.2.2, -m genome, Bapsfontein Gauteng, South Africa) [29]. BUSCO was used to identify the conserved genes of the “eukaryota_odb10”, “insecta_odb10”, and “diptera_odb10” databases in the produced assemblies. BUSCO (v5.2.2, -m proteins) was also used to assess the completeness of gene predictions (see “Genome annotation” paragraph) against the “insecta_odb10” and “diptera_odb10” databases. Furthermore, BUSCO (v5.2.2, -m genome) was used to assess the completeness of the soft-masked final assembly (see “Genome annotation” paragraph) and the previous *H. illucens* reference genome (GCA_905115235.1 assembly) against “insecta_odb10” and “diptera_odb10” databases.

### 2.5. Genome Visualization and Analysis of the Chromatin Organization

A Hi-C contact map of the final assembly was generated and visualized in HiGlass, a web-based viewer for exploring and comparing genomic contact matrices, defining which regions of a genome are physically close together and navigating across genomic loci at different resolutions [30]. TADs were identified using the Arrowhead program implemented in the Juicertools package (the parameters used for predicting TADs at 10, 25, and 50 kbp resolution were -k KR -m 2000 -r 10000, -k KR -m 2000 -r 25000, and -k KR -m 2000 -r 50000, respectively) [31], A/B compartments using the eigenvector program implemented in the Juicertools package (parameters: KR BP 1000000), CTCF binding sites using the cread program (the position weight matrix was downloaded from the CTCFBSDB 2.0 website, https://insulatordb.uthsc.edu, accessed on 31 January 2022), and isochores using the isofinder program (parameters: 0.90 p2 3000) [32].

### 2.6. Genome Annotation

The de novo gene prediction on the final assembly was performed using AUGUSTUS (v3.4.0) [33]. Coding sequences and peptide sequences from *H. illucens* were used to train the initial ab initio model for *H. illucens* using the web AUGUSTUS Service [34]. AUGUSTUS (v3.4.0) was then used to predict genes in the final assembly [35]. To identify and quantify the repetitive regions within the BSF genome, Dovetail Genomics modeled a de novo library of repetitive sequences using RepeatModeler (v2.0.1) [36]. The custom repeat library obtained from the RepeatModeler library was used to identify soft mask repeats in the final assembly using RepeatMasker (v.4.1.0.) [37]. tRNAs were predicted using the software tRNAscan-SE (version 2.0.5) [38]. Coding sequences from *Aedes aegypti* (Diptera Culicidae), *H. illucens*, and *Lucilia cuprina* (Diptera Calliphoridae) were also used to train the initial ab initio model for *H. illucens* using the AUGUSTUS software (v2.5.5). Six rounds of prediction optimization were conducted with the software package provided by AUGUSTUS. The same coding sequences were also used to train a separate ab initio model for *Hermetia illucens* using SNAP (v2006-07-28) [39]. Published RNA-seq reads obtained from whole larva (study accession: PRJEB19091) [40] were mapped onto the genome using the STAR aligner software (v2.7) [41] and intron hints generated with the bam2hints tools within the AUGUSTUS software. MAKER [42], SNAP, and AUGUSTUS (with intron–exon boundary hints provided from RNA-Seq) were then used to predict genes in the repeat-masked reference genome. Swiss-Prot peptide sequences from the UniProt database (http://www.uniprot.org/uniprot, accessed on 30 May 2022) [43,44] were downloaded and used in conjunction with the protein sequences from *A. aegypti*, *H. illucens*, and *L. cuprina* to generate peptide evidence in the Maker pipeline. Only genes that were predicted using both the SNAP and AUGUSTUS software were retained in the final gene sets. To help assess the quality of the gene prediction, annotation edit distance (AED) scores were generated for each of the predicted genes as part of the MAKER pipeline. The AED score is a general measure of how well the predicted gene is supported by external evidence. The AED score ranges from 0 to 1, and a lower score represents a stronger supporting evidence for the gene. Genes were further characterized for their putative function by performing a BLAST search of the peptide sequences against the UniProt database. The *hiLsp-2* gene was further manually curated using publicly available full-length transcript evidence as a guide. The conserved domains of *hiLsp-2* were defined by means of a search against the database CDD v3.19–58235 PSSMs using CD-Search, a web-based tool for the detection of structural and functional domains in protein sequences [45]. Sequence alignment analyses were performed with blast [46]. The proteins’ 3D structures were predicted with AlphaFold [47] by using the ColabFold platform (https://github.com/YoshitakaMo/localcolabfold, accessed on 13 September 2022) [48]. PDB files generated by ColabFold were then visualized and aligned with each other by using the iCn3D Structure Viewer (https://www.ncbi.nlm.nih.gov/Structure/icn3d/, accessed on 13 September 2022) [49,50].

## 3. Results

### 3.1. Hermetia Illucens Rearing and Crossing

For the *H. illucens* genome sequencing and assembly, a partial inbreeding procedure was started to increase the homozygosity in the population. For that, a couple of adults (1 ♂ and 1♀) were isolated in a cage to allow for reproduction with the same methodology described for *H. illucens* mass rearing. Once the F3 adult generation was obtained, ten unsexed adults were degutted with microsurgery scissors to remove the symbiotic bacteria. Then, all samples were stored at −80 °C and sent to Dovetail Genomics laboratories for genome sequencing and assembly.

### 3.2. Genome Library Construction and Sequencing and Genome Assembly

The PacBio CCS sequencing library contained 1,210,120 reads. A total of 16.3 gigabases (Gb) was generated, providing a 14-fold coverage based on the 1.2 Gb estimated genome size. The Omni-C library was sequenced on an Illumina HiSeqX platform to produce an approximately 30x sequence coverage. Using the de novo assembler Hifiasm on the PacBio data, an initial draft assembly with a size of 1.31 Gb, a scaffold N50 value of 3.79 Mb, and a scaffold L50 value of 78 was assembled (Table 1). Following polishing with blobtools and purge_dups, the resulting primary filtered assembly had a size of 888.5 Mb, a scaffold N50 value of 5.34 Mb, and a scaffold L50 value of 44 (Table 1). For scaffolding the final assembly using the primary filtered assembly as the input assembly and HiRise as scaffolding software (version 2.0.5), 128,145,702 read-pairs of the Omni-C library were used. No breaks were made to the primary filtered assembly by HiRise. After the second round of scaffolding, the resulting chromosome-level final assembly consisted of a total length of 888.59 Mb, with a scaffold N50 value of 162.19 Mb and a scaffold L50 value of 3 (Table 1). Although the final assembly was organized in 169 scaffolds, 98% of the assembly (878 Mb) was contained in seven scaffolds, which corresponds to the haploid number of chromosomes for this species.

### 3.3. Genome Quality Evaluation

The initial draft assembly, the primary filtered assembly, and the final assembly covered 99.2%, 88.9%, and 89.1% of the Insecta BUSCO (v5.2.2) core genes, respectively, and 95.6%, 84.6%, and 84.7% of the Diptera BUSCO core genes, respectively (Figure 2). The percentages described above were calculated as the fraction of complete single-copy or duplicated genes relative to the entire gene dataset. Particularly, among the core genes of Insecta, 87.9% of genes were identified in the final assembly as a single copy (S), 1.2% as duplicated (D), 1.1% of genes as fragmented (F), and 9.8% of genes as missing (M) (Figure 2); among the core genes of Diptera, 83.7% of genes were identified in the final assembly as a single copy (S), 1.0% of genes as duplicated (D), 1.6% of genes as fragmented (F), and 13.7% of genes as missing (M) (Figure 2).

In the initial draft assembly, blobtools identified 77.90% (1021.35 Mb) of the raw PacBio data as exclusively arthropod sequences (Figure 3). Segments of the closest sequence similarity to a virus (12.48%), Annelida (0.42%), Chordata (0.09%), Mollusca (0.01%), Nematoda (0.009%), and Cnidaria (0.001%) were also identified. However, 119.07 Mb (9.08%) showed no significant taxonomic identification (Figure 3). Scaffolds identified as possible contaminants and sequences with no significant taxonomic identification were removed from the primary filtered assembly, which, therefore, did not include any significant contaminant sequences. The BUSCO statistics were also calculated for the blobtools-filtered draft assembly (see Appendix A): this assembly covered 89.5% and 85.2% of the Insecta and Diptera BUSCO core genes, respectively. Among the core genes of Insecta covered by the blobtools-filtered draft assembly, 70.3% of the genes were a single copy (S) and 9.5% of the genes were missing (M); among the core genes of Diptera covered by the final assembly, 67.5% of the genes were a single copy (S) and 13.2% of the genes were missing (M).

### 3.4. Genome Visualization and Analysis of the Chromatin Organization

The chromosome organization of the HiRise assembly was visualized by the heatmap generated with HiGlass, which identified seven chromosomes (Figure 4). The topologically associated domains (TADs) analysis identified 635, 630, and 395 TADs at 10 kbp, 25 kbp, and 50 kbp resolution, respectively. A total of 65, 65, 7671, and 0 A compartments, B compartments, CTCF binding sites, and Isochores were predicted, respectively.

### 3.5. Genome Annotation

The de novo gene prediction performed with AUGUSTUS on the final assembly annotated 54.409 genes which provided BUSCO completeness scores of 85.1% and 79.5% for Insecta and Diptera core gene datasets, respectively. Repeat masking performed by Dovetail Genomics resulted in a total of 65.62% of the final assembly being identified as repeat sequences. Class I transposable elements (TEs) repeats (46.38%) were the most abundant class of repetitive elements identified, while Class II TEs repeats covered 3.14% of the final assembly. The soft-masked final assembly covered 89.1% and 84.7% of the Insecta and Diptera BUSCO core genes, respectively. Among the core genes of Insecta, 87.9% of genes were identified as a single copy (S) in the soft-masked final assembly, 1.2% of genes as duplicated (D), 1.1% of genes as fragmented (F) and 9.8% of genes as missing (M); among the core genes of Diptera, 83.7% of genes were identified as a single copy (S), 1.0% of genes as duplicated (D), 1.6% of genes as fragmented (F), and 13.7% of genes as missing (M). Comparing the BUSCO results between our soft-masked final assembly and the previous *H. illucens* reference genome (GCA_905115235.1 assembly) (Figure 5), 135 Insecta BUSCO genes were missing from our assembly, whereas 13 Insecta BUSCO genes were missing in the GCA_905115235.1 assembly. The *H. illucens* reference genome also contains a higher number of Insecta single-copy orthologous genes than our final assembly (1344 and 1201 single-copy genes, respectively), while the gene duplication is low both in the reference genome and in our soft-masked final assembly (5 and 16 duplicated genes, respectively).

A gene prediction was also performed on the soft-masked final masked assembly using the maker pipeline. This annotation pipeline annotated 32.516 genes (Table 2), which provided BUSCO completeness scores of 84.6% and 76.3% against Insecta and Diptera core genes datasets, respectively (Figure 6). Among the core genes of Insecta, 84.3% of genes were identified by the gene prediction as a single copy (S), 0.3% of genes as duplicated (D), 2.9% of genes as fragmented (F), and 12.5% of genes as missing (M) (Figure 6); among the core genes of Diptera, 75.8% of genes were identified through the gene prediction as a single copy (S), 0.5% of genes as duplicated (D), 5.4% of genes as fragmented (F), and 18.3% of genes as missing (M) (Figure 6).

### 3.6. Lsp-2 Annotation

Among the genes predicted by Dovetail Genomics using the MAKER pipeline, two (*ANN16409* and *ANN16413*) were found to have similarities with the unique *dmeLsp-2* gene. The alignment analyses between the proteins encoded by *ANN16409* and *ANN16413* with the protein isoforms encoded by *dmeLsp-2* showed alignments with a particularly high query coverage (99%), despite the percentage identity being less than 50%. At the transcript level, the query coverage was always less than 30%, with a maximum identity rate of 66%. *ANN16409* and *ANN16413* had both a really low AED score (0.07 and 0.00, respectively), indicating strong supporting evidence. Both *ANN16409* and *ANN16413* were located on the negative strand of chromosome 3 at a distance of approximately 69 kb from each other. Moreover, both genes were single-exon genes. *ANN16409* encoded for a single transcript of 2136 bp, and *ANN16413* encoded for a single transcript of 2139 bp. *ANN16409* was also manually curated by Dovetail Genomics (Figure 7).

From the alignment analysis performed with blast, the transcript of *ANN16409* aligned against the transcript XM_038054670 of *LOC119651216* (Gene ID: 119651216), a gene described in the *H. illucens* reference genome as larval serum protein 2-like, with a query coverage of 89% and an identity percentage of 100%. Particularly, the region with perfect alignment corresponded to the entire CDS of *LOC119651216*. The transcript of *ANN16413* aligned against the transcript XM_038055158.1 of *LOC119651526* (Gene ID:119651526), respectively, another gene described in the *H. illucens* reference genome as larval serum protein 2-like, with a query coverage of 95% and an identity percentage of 100%. Again, the region with perfect alignment corresponded to the entire CDS of *LOC119651526*. These alignments were also conserved at the protein level: the protein encoded by *ANN16409* and that encoded by *ANN16413* aligned against the protein encoded by *LOC119651216* and *LOC119651526*, respectively, both with a query coverage and an identity percentage of 100%. The alignment of the protein encoded by *ANN16409* with that encoded by *ANN16413* revealed significant differences between these two protein structures: the two proteins have 218 amino acid substitutions relative to each other; of these, 93 are non-conservative substitutions involving amino acids with different chemical properties or substitutions that are not frequently observed in related proteins. Furthermore, in order to identify conserved domains in these proteins, the NCBI Conserved Domains database was interrogated with the corresponding sequences, protein, and transcript of *ANN16409* and *ANN16413* genes, respectively. The results showed that both *ANN16409* and *ANN16413* had three conserved domains as Hemocyanin_C (pfam03723), Hemocyanin_M (pfam00372), and Hemocyanin_N (pfam03722), typical of arthropod hemocyanins and insect larval storage proteins. The three-dimensional organization of the two proteins encoded by *ANN16409* and *ANN16413* was predicted with Alphafold and visualized with iCn3D Structure Viewer, as shown in Figure 8.

The 3D protein structures of *ANN16409* and *ANN16413* were also compared with each other and with that reported for *dmeLsp-2*, recovered from UniProt (Figure 9).

## 4. Discussion

### 4.1. Comparative Genome Assembly Analysis

Comparing the statistics of the final assembly with the statistics of the other draft assemblies, as expected, the final assembly was assembled in a lower number of scaffolds and had a higher N50 value, indicative of a higher contiguity quality (Table 1). The final assembly is smaller in size than the reference genome (GCA_905115235.1), 888 Mb and 1.01 Gb, respectively. This is probably due to the identification of more contaminating sequences by blobtools, considering that the size of the initial draft assembly was 1.3 Gb, whereas the genome size of the first draft assembly of the reference genome was reported as 1.09 Gb [18]. Our final assembly is also assembled in a higher number of scaffolds (169 and 20, respectively) and has a lower N50 value (162.19 Mb and 180.36, respectively) than the reference genome, indicative of lower quality in the assembly process (Table 3). These results were, however, to a certain extent expected, considering that for the reference genome, two different PacBio libraries, a 10x Genomics Chromium linked read 150 bp PE library and a Hi-C PE library, were prepared and sequenced [18]. However, excluding the reference genome and comparing the statistics of our final assembly with those generated by Generalovic et al. [18] for publicly available assemblies of other species of Diptera, our final assembly is the highest-quality assembled dipteran genome available amongst those sampled, assembled into the smallest number of scaffolds with the largest N50 value (Table 3).

Comparing the BUSCO scores of the different assemblies produced by Dovetail Genomics, it was observed that after the step performed with blobtools, the number of complete genes (C) reduced while the number of missing genes (M) increased compared to the first draft assembly (Figure 2). This was probably because some of the conserved genes not identified after running blobtools were contained in the scaffolds and considered as possible contaminants. As expected, in the primary filtered and final assemblies, the number of single-copy orthologs (S) increased, and the number of duplicated genes (D) reduced compared to the initial draft assembly. The higher number of single-copy orthologs and the reduction of duplicated genes in the primary filtered and final assemblies should be due to the removal of haplotigs and contig overlaps using purge_dups and to the introduction of the Omni-C reads. The BUSCO scores showed some differences between the existing high-quality Diptera genomes sampled by Generalovic et al. (BUSCO score generated by Generalovic et al. with BUSCO v3.0.2 and “insecta_odb9” database) and our assemblies (BUSCO score generated with BUSCO v5.2.2 and “insecta_odb10” database (Table 3). For the blobtools-filtered assembly, the primary filtered assembly and the final assembly, the percentage of core genes (C) identified by BUSCO was reduced compared to that defined for the reference genome assembly and other publicly available assemblies of other Diptera species (see “Genome quality evaluation” paragraph and Table 3). Blobtools-filtered, primary filtered, and final assemblies also had a higher percentage of missing genes (M). This could be partly explained by the removal of sequences identified as possible contaminants with blobtools. Indeed, in the first draft assembly, the percentage of conserved genes (C) and of missing genes (M) defined by BUSCO was similar to that reported in the reference genome assembly and other publicly available assemblies of other Diptera species, although a higher percentage of duplicated genes was defined.

### 4.2. Structural Organization of the Genome

As expected, the Higlass visualization of the final assembly recognized seven chromosomes, identifiable as large squares (Figure 3). Of particular interest was chromosome 7, previously identified by Generalovic et al. as a sex chromosome, due to its low autosomal coverage in males [18]. The length of chromosome 7 defined in this de novo genome assembly was also reminiscent of *D. melanogaster* chromosome 4. However, chromosome 7 of *H. illucens* was previously defined as a non-redundant sex chromosome, while chromosome 4 of *D. melanogaster*, also known as dot chromosome, is an ancestral X chromosome, then reverted to an autosomal chromosome, as showed by Vicoso and Bachtrog [51]. The identification of zero isochores by using the isofinder program was of interest. Isochores are extended genomic regions (typically 300 kb to multimegabase) of uniform, characteristic GC content, generally identifiable in metazoan genomes [52]. The absence of isochores in our final assembly was particularly interesting. This could be due to various environmental factors that play a role in determining genome organization and chromatin structure, such as the body temperature [52], isochore erosion by G/C to A/T substitution dominance [53], and the fact that the heterogeneity in GC content seems to be much less evident in insects [54]. This may have made identification of the isochores more difficult, especially for reduced-size isochores [53]. The non-identification of isochores could, however, also be due to technical reasons, such as the procedure used in assessing isochores. For this purpose, the process of identifying isochores would be further investigated, which could lead to changes in the number of isochores identified.

### 4.3. Genome Annotation

The de novo gene prediction performed with AUGUSTUS on the final assembly annotated 54.409 genes which provided BUSCO completeness scores of 85.1% and 79.5% for Insecta and Diptera core gene datasets, respectively. The number of genes predicted with AUGUSTUS on the final assembly was significantly higher than the number of genes reported in the reference genome on the UCSC Genome Browser (54,409 vs. 22,369 genes, respectively). The very high number of predicted genes in the final assembly may depend on having used only AUGUSTUS as ab initio gene prediction software and on not using RNA-Seq data as extrinsic evidence of gene structures. The repeat masking performed with RepeatMasker using the custom repeat library obtained from RepeatModeler identified 65.62% of the final assembly as repeat sequences, a very close percentage to that reported in the current reference genome (67.32%) [18]. The soft-masked assembly also had the same BUSCO statistics as the final assembly, against both the Insecta and Diptera conserved gene datasets. Taken together, these data indicate that the repeat masking process occurred correctly and that it did not lead to masking even BUSCO-conserved genes. Comparing the BUSCO results between our soft-masked final assembly and the previous *H. illucens* reference genome (GCA_905115235.1 assembly) (Figure 5), 135 Insecta BUSCO genes were missing from our assembly, whereas 13 Insecta BUSCO genes were missing in the GCA_905115235.1 assembly. The *H. illucens* reference genome also contains a higher number of Insecta single-copy orthologous genes than our final assembly (1344 and 1201 single-copy genes, respectively), while the gene duplication is low both in the reference genome and in our soft-masked final assembly (5 and 16 duplicated genes, respectively). We believe that the increased number of missing genes and the reduced number of single-copy orthologous genes in our soft-masked final assembly depended on the genome assembly step performed with blobtools, which may have removed scaffolds containing some of the conserved genes from the BUSCO datasets. The gene prediction performed by Dovetail Genomics on the soft-masked final assembly using the MAKER pipeline identified 32,516 genes. This number of predicted genes was higher than that reported in the reference genome (22,369). This was probably because each predicted gene was associated with a single transcript. In this way, individual genes coding for several alternative transcripts may have been predicted as separate genes. This represents an important limitation in comparing annotations across genomes. At the same time, it still constitutes a crucial initial step in genomic annotation that will be further expanded to achieve a more accurate representation of the exact number of genes. The BUSCO analysis of the gene prediction performed by Dovetail Genomics indicated a not-too-high percentage of single-copy genes, especially among the conserved gene of Diptera. These BUSCO percentages do not indicate excessively high completeness for gene prediction. We believe that this is not due to the low accuracy of the procedure used for gene prediction, but it is rather due to the removal of some scaffolds during the assembly procedure performed with blobtools, as described above. The manual annotation phase recently started and it could lead to a change in the number of annotated genes in our soft-masked final assembly and the improvement of BUSCO statistics on gene predictions.

### 4.4. Lsp-2

Considering future genetic engineering applications aimed at improving the nutritional properties of *H. illucens*, the identification and characterization of endogenous genes to be used as targets for future site-specific genome editing strategies is of particular interest. Due to its observed temporal expression profile in *D. melanogaster*, the *Lsp-2* gene emerges as an interesting target among such candidate genes [55]. Two predicted genes (*ANN16409* and *ANN16413*) with significant similarity to *dmeLsp-2* were found. However, while *D. melanogaster* possesses only one copy of *Lsp-2*, in both the reference genome and the final assembly we annotated, the presence of two *Lsp-2* genes, clustered at a single genomic site, was highlighted. This could indicate the formation of a multigene family from the single ancestor gene of *Lsp-2*, which occurred after Muscomorpha and Stratiomyomorpha diverged. This might resemble what has already been observed for the *Lsp-1*/arylphorin hexamerin group, whereby a different gene copy number was observed between the Acalyptratae and the Calyptratae [56]. The presence of two gene copies of *hiLsp-2* could be due to a number of reasons: one of the two gene copies could actually be a pseudogene, but *ANN16409* and *ANN16413* could also be expressed at different stages of the *H. illucens* life cycle. Moreover, in *D. melanogaster*, *Lsp-2* encodes the single subunit of an homoexameric larval serum protein with a molecular weight of about 450-kDa, composed of glycosylated subunits with a molecular weight of about 74.5-kDa [57,58]. Thus, we also cannot exclude the possibility that in *H. illucens*, *Lsp-2* is organized not as a homohexamer but rather as a heterohexamer consisting of subunits of *ANN16409* and *ANN16413*, or that *ANN16409* and *ANN16413* assemble into two different homohexamerins. Molecular analyses of *hiLsp-2* may, in the future, shed light on the biological role of these two gene copies. *ANN16409* and *ANN16413* localized on chromosome 3, as the *hiLsp-2* genes were reported in the reference genome (*LOC119651526* and *LOC119651216*). *ANN16409*, *ANN16413*, and the *hiLsp-2* genes were all single-exon genes. This gene structure is consistent with that observed for the *dmeLsp-2* gene, revealing a similar gene structure in both species. Assessing the length of the transcripts associated with these genes, they were characterized by a length of between 2136 and 2477 bp, while the length of the translated proteins was between 701 and 713 aa. From the conserved domain analysis, as expected, ANN16409 and *ANN16413* had three hemocyanin conserved domains by the presumed functionality of insect larval storage protein associated with these two genes. From the alignment analysis performed with blast, we believe that the first *hiLsp-2* gene annotated by us (*ANN16409*) corresponds to the gene *LOC119651216* reported in the reference genome, while the second *hiLsp-2* gene annotated by us (*ANN16409*) corresponds to the gene *LOC119651526* reported in the reference genome. Indeed, as for *ANN16409* and *ANN16413*, *LOC119651526* and *LOC119651216* were also 69 kbp apart on the reference genome chromosome 3. From the structural alignment analysis (Figure 9), we conclude that *ANN16409* and *ANN16413* have the same protein fold and that both have a high structural similarity with *dmeLsp-2*. Particularly, the TM-score was always found to be above 0.9, while TM-score >0.5 is generally considered the threshold value for proteins with the same folding [59]. These results indicate that despite the important differences in primary structure between *ANN16409* and *ANN16413* (218 amino acid substitutions, of which 93 are non-conservative concerning each other), greater structural, and probably functional, conservation is maintained.

## 5. Conclusions

We assembled a chromosome-scale genome for the *H. illucens* population reared at DAFE, by combining PacBio and Omni-C proximity ligation technology. The final assembly was 888.59 Mb in size, with a scaffold N50 value of 162.19 Mb and a scaffold L50 value of 3. These statistics indicated a lower assembly quality than that reported in the reference genome, probably due to the preparation and sequencing of fewer libraries. However, our *H. illucens* final assembly was found to be one of the dipterans’ highest-quality genome assembly, comparing it with the statistics reported for those of other dipteran species. Moreover, we performed a chromosome-scale analysis of the structural organization of the *H. illucens* genome, which allowed us to define TADs and other topological features. In a chromatin context, these topological features are extremely important for the regulation of gene expression. The acquisition of such data will, therefore, make it possible to increase knowledge about gene regulation in *H. illucens*. We also annotated 32,516 genes using the MAKER pipeline. Among them, we continued the annotation of the *hiLsp-2* gene, for which two gene copies were identified, and we predicted the three-dimensional organization of the putative proteins codified by these gene copies using Alphafold. The alignment analysis of the three-dimensional structures showed that the proteins derived from the *Lsp-2* gene copies have the same protein fold and that both have a high structural similarity with *dmeLsp-2*, probably indicating functional conservation. The availability of a chromosome-scale assemblage for the *H. illucens* population reared at the DAFE, combined with the annotation and characterization of specific genes, may help develop genome editing strategies for this insect species of great biotechnological interest.

## Figures and Tables

**Figure 1 insects-15-00133-f001:**
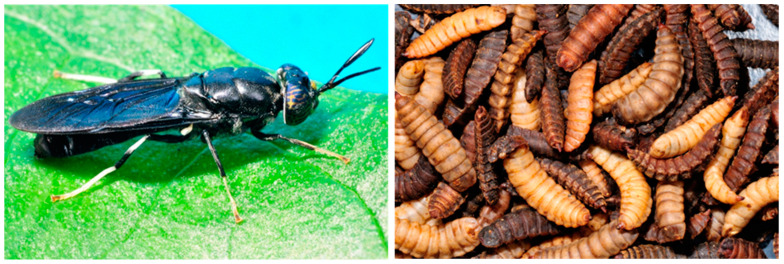
*Hermetia illucens* adult (**left**); larvae and pupae (**right**).

**Figure 2 insects-15-00133-f002:**
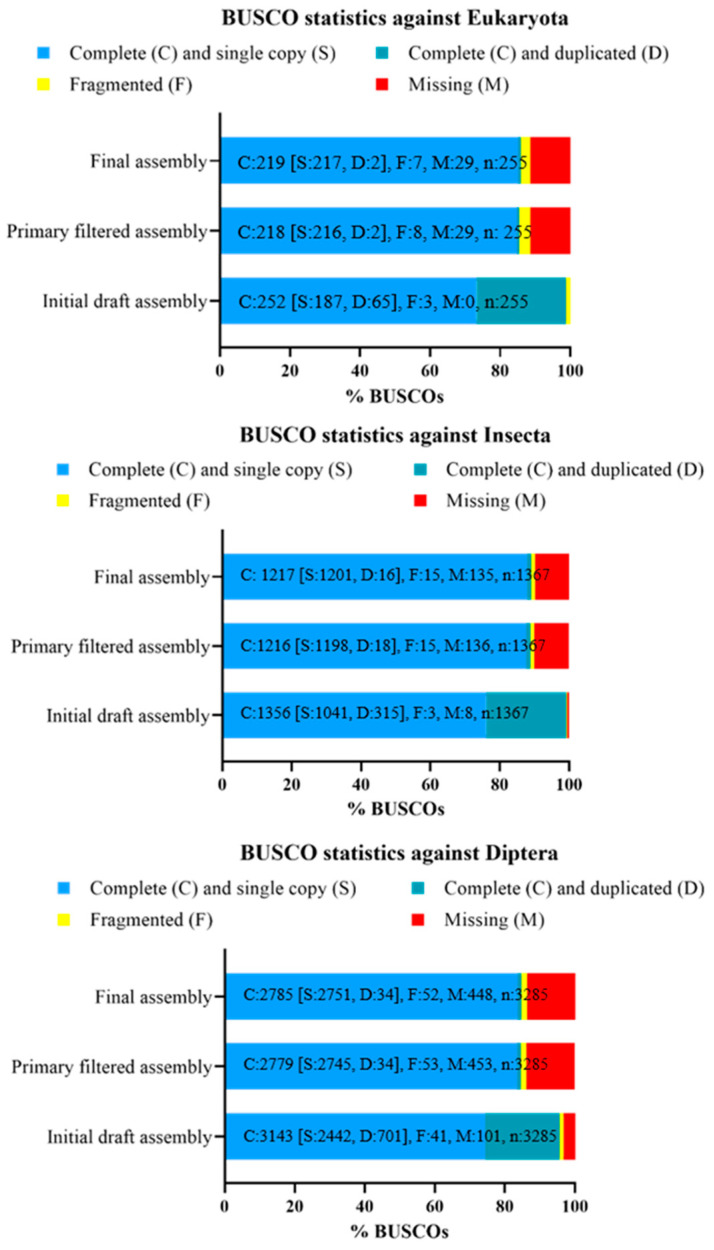
BUSCO scores of our *Hermetia illucens* assemblies generated from different databases. The initial draft assembly presented a reduced number of missing genes (M) compared to the subsequent draft and final assemblies. Databases used to assess completeness against eukaryotes, insects, and dipterans: “Eukaryote_odb10”, “insecta_odb10”, and “diptera_odb10” databases, respectively. BUSCO = Benchmarking Universal Single-Copy Orthologs.

**Figure 3 insects-15-00133-f003:**
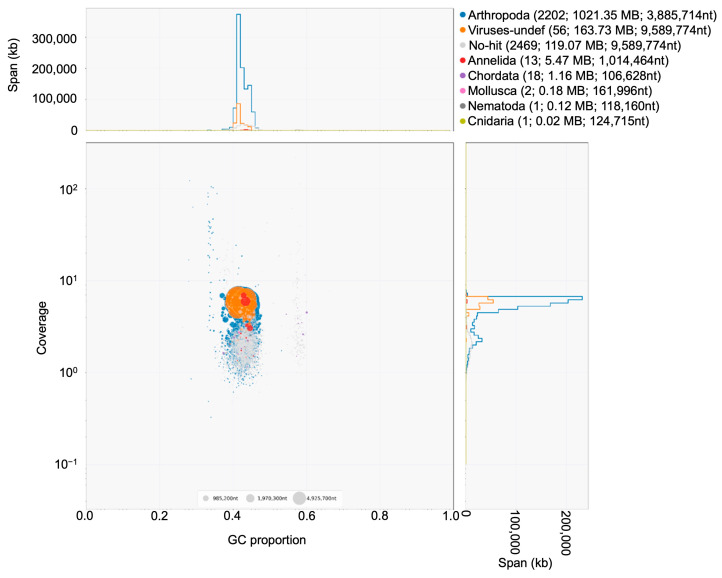
Taxon-annotated GC-coverage plot of the initial draft assembly. Each scaffold or contig is represented in the scatter plot by a single filled circle with a diameter proportional to the sequence length and with a color specific to the taxonomic affiliation. Each circle is placed on the X-axis based on its GC proportion and on the Y-axis based on the base coverage of the sequence in the coverage library. The legend in the top right-hand corner shows the number of scaffolds or contig, total span, and N50 of the sequences belonging to each taxonomic group. nt = nucleotides, Kb = kilobase, plot generated using blobtools (v1.1.1).

**Figure 4 insects-15-00133-f004:**
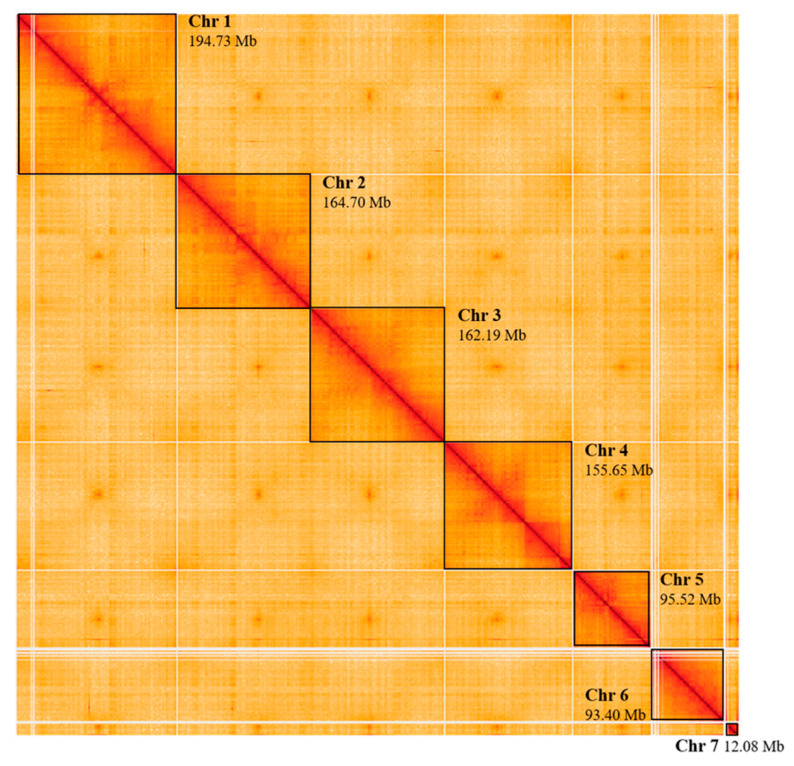
HiGlass contact map of chromosomal interactions. In the contact map, each point represents how often two regions of the genome were found to be close enough together to be ligated, and regions of the map associated with more intense color represent genomic regions with the greater physical association. In this representation, chromosomes are visible as large squares and are highlighted here with black frames. The length of each chromosome is shown in the contact map above each of them. Mb = megabase.

**Figure 5 insects-15-00133-f005:**
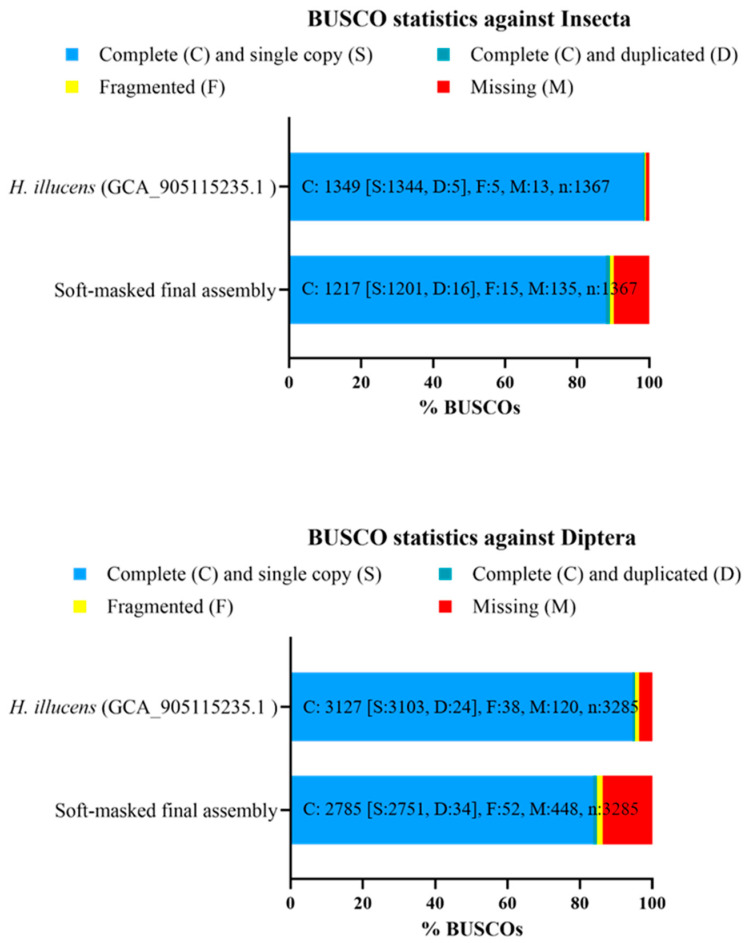
BUSCO score of *Hermetia illucens* reference genome and our soft-masked final assembly. Scores were generated from the “insecta_odb10” and “diptera_odb10” datasets. BUSCO = Benchmarking Universal Single-Copy Orthologs.

**Figure 6 insects-15-00133-f006:**
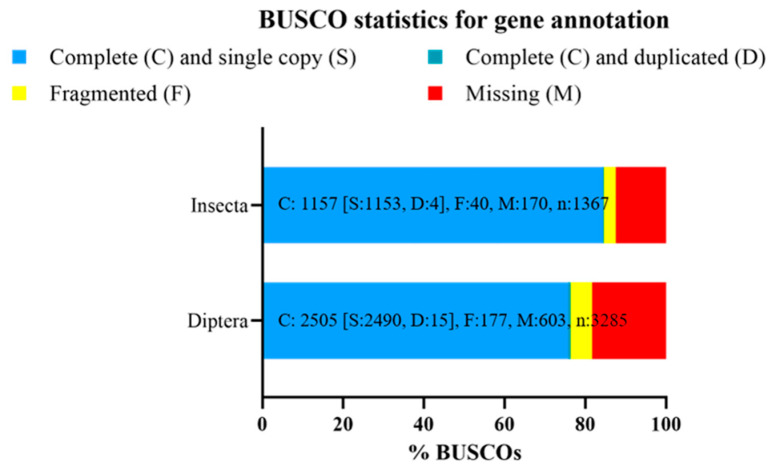
BUSCO score of the gene annotation made on the soft-masked final assembly. The gene annotation missed the prediction of several conserved genes (12.5 and 18.3% of Insecta and Diptera conserved genes, respectively). Databases used to assess gene annotation completeness against insects and dipterans: “insecta_odb10” and “diptera_odb10” databases, respectively. BUSCO = Benchmarking Universal Single-Copy Orthologs.

**Figure 7 insects-15-00133-f007:**
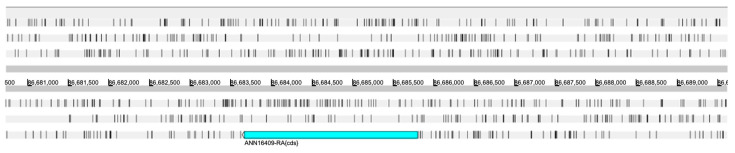
Visualization of gene structure for *ANN16409*. The light blue rectangle represents the CDS of *ANN16409* organized into a single exon. The white arrowhead on the left of the light blue rectangle (CDS) represents the gene orientation.

**Figure 8 insects-15-00133-f008:**
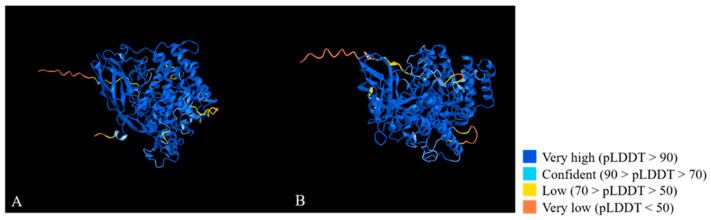
The 3D protein structure prediction of *ANN16409* (**A**) and *ANN16413* (**B**). The color representation of each amino acid depends on the accuracy of its prediction within the protein structure according to the legend on the right. pLDDT = predicted local distance difference test.

**Figure 9 insects-15-00133-f009:**
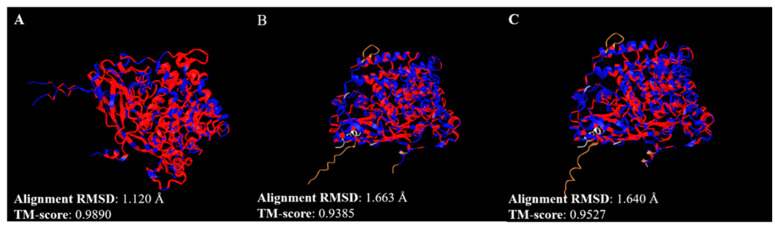
Structural alignment of *ANN16409* and *ANN16413* (**A**), *ANN16409* and *dmeLsp-2* (**B**), and *ANN16413* and *dmeLsp*-2. The structural alignments were created on iCn3D Structure Viewer, using TM-align as the algorithm for protein structure comparisons. RMSD = root mean square deviation, TM-score = template modelling score. In these representations, all matching chains are superposed, while the aligned residues are colored in red for identical residues or blue for non-identical residues. In panel B, non-aligned residue of *dmeLsp-2* and *ANN16409* are colored light grey and brown, respectively; in panel (**C**), non-aligned residue of *dmeLsp-2* and *ANN16413* are colored light grey and brown, respectively.

**Table 1 insects-15-00133-t001:** Assembly statistics of our *Hermetia illucens* draft and final assemblies. bp = base pair, Mb = megabase.

Assembly	Total Length (bp)	N50 (Mb)	L50	Scaffold Number
Initial draft assembly	1,311,107,399	3,793,743	78	4762
Primary filtered assembly	888,493,946	5,340,478	44	1185
Final Assembly	888,595,941	162,194,137	3	169

**Table 2 insects-15-00133-t002:** Genome annotation statistics of our *Hermetia illucens* soft-masked final assembly. bp = basepair.

Genome Annotation Statistics	Number
Total number of genes	32,516
Total coding region (bp)	36,672,320
Average length of genes (bp)	1127.82
Number of single-exon genes	2091

**Table 3 insects-15-00133-t003:** Assembly statistics and BUSCO score of our *Hermetia illucens* final assembly and selection of Diptera genomes. The assembly statistics and the BUSCO scores of the selection of Diptera genomes were made by Generalovic et al. BUSCO scores of our *Hermetia illucens* final assembly were generated from the “insect_odb10” database, and BUSCO scores calculated by Generalovic et al. were generated from the “insecta_odb9” database. BUSCO = Benchmarking Universal Single-Copy Orthologs, Mb = megabase, C = complete genes, S = complete and single copy genes, D = complete and duplicated genes, F = fragmented genes, M = missing genes.

Species Name	Scaffold Number	N50 Value (Mb)	BUSCO %				
			C	S	D	F	M
Final assembly	169	162.19	89.1	87.0	1.2	1.1	9.8
*Hermetia illucens* (reference genome)	20	180.36	98.6	97.8	0.8	0.5	0.9
*Hermetia illucens* (GCA_009835165.1)	2806	1.70	98.9	91.1	7.8	0.6	0.5
*Drosophila melanogaster* (GCA_000001215.4)	1870	25.29	99.7	99.0	0.7	0.2	0.1
*Drosophila virilis* (GCA_000005245.1)	13,530	31.08	99.1	98.1	1.0	0.4	0.5
*Musca domestica* (GCA_000371365.1)	20,487	0.23	98.6	96.9	1.7	0.4	1.0
*Stomoxys calcitrans* (GCA_001015335.1)	12,042	0.50	98.4	97.7	0.7	1.0	0.5
*Glossina morsitans* (GCA_001077435.1)	24,071	/	98.9	96.6	2.3	0.6	0.5
*Aedes aegypti* (GCA_002204515.1)	2310	0.41	98.9	94.5	4.4	0.4	0.7
*Culex quinquefasciatus* (GCA_000209185.1)	3171	0.49	96.7	91.8	4.9	0.8	2.5

## Data Availability

The datasets generated and analyzed during the current study are available in the European Nucleotide Archive (ENA) repository. Study Accession Number: PRJEB58627. Sample Accession Number: ERS14372541.

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
