# Peer review of "De Novo Genome Assembly at Chromosome-Scale of Hermetia illucens (Diptera Stratiomyidae) via PacBio and Omni-C Proximity Ligation Technology"

_insects, 2024, doi:10.3390/insects15020133_

Round 1

Reviewer 1 Report

Comments and Suggestions for Authors

Hermetia illucens, also known as Black Soldier Fly (BSF), has attracted the attention of researchers thanks to its potential use, at industrial scale, as bio-converter,  food  and anti-microbial peptide producer, among other possibilities.  Given the increasing interest by this dipteran, the authors introduce in this manuscript an additional genome assembling to be added to three previous papers on the same subject. Although the work is not exactly original, by expanding the genome knowledge of this species is a welcome contribution specially when putative biotechnological applications using this fly are implied. 

- An information that I have missed is concerned with the  "natural history" of the fly, that could make a contrast with genomic data. I think the authors are able to produce a figure containing images of adult as well as immature stages in order to give a better idea of black soldier flies to the reader. Ideally, the figure might also include larvae feeding in their environment.

-In relation the chromosome-scale genome of H. illucens presented by the authors, there is an interesting result regarding what would be called chromosome 7: its length is reminiscent of Drosophila melanogaster chromosome 4 that has exceptional features when compared to the rest of the fruit-fly chromosome complement. I ask the authors if there are published data of mitotic chromosomes of this fly in order to be compared with the genome assembly obtained in this work. Alternatively, if they can perform chromosome spreads for such a figure. I think my above observations meet the author's interest for chromatin structure as stated in the text and would therefore enrich the manuscript with relevant information.

Author Response

Comments and Suggestions for Authors

Hermetia illucens, also known as Black Soldier Fly (BSF), has attracted the attention of researchers thanks to its potential use, at industrial scale, as bio-converter, food  and anti-microbial peptide producer, among other possibilities.  Given the increasing interest by this dipteran, the authors introduce in this manuscript an additional genome assembling to be added to three previous papers on the same subject. Although the work is not exactly original, by expanding the genome knowledge of this species is a welcome contribution specially when putative biotechnological applications using this fly are implied.

- An information that I have missed is concerned with the  "natural history" of the fly, that could make a contrast with genomic data. I think the authors are able to produce a figure containing images of adult as well as immature stages in order to give a better idea of black soldier flies to the reader. Ideally, the figure might also include larvae feeding in their environment.

R: We are very grateful to the reviewer for this request, which we are happy to fulfil by including a figure with pictures of the species.

-In relation the chromosome-scale genome of H. illucens presented by the authors, there is an interesting result regarding what would be called chromosome 7: its length is reminiscent of Drosophila melanogaster chromosome 4 that has exceptional features when compared to the rest of the fruit-fly chromosome complement. I ask the authors if there are published data of mitotic chromosomes of this fly in order to be compared with the genome assembly obtained in this work. Alternatively, if they can perform chromosome spreads for such a figure. I think my above observations meet the author's interest for chromatin structure as stated in the text and would therefore enrich the manuscript with relevant information

R: We thank the reviewer for raising this interesting point regarding the comparison between the chromosome 7 of H. illucens and the chromosome 4 of D. melanogaster. According to the referee’s suggestion, we searched for further information about H. illucens chromosome 7 and D. melanogaster chromosome 4 and we added such literature findings in the discussion section of the paper.  The “Structural organization of the genome” appears as it follows in the revised version of the manuscript.

As expected, Higlass visualization of the final assembly recognized seven chromosomes, identifiable as large squares (Figure 3). Of particular interest was chromosome 7, previously identified by Generalovic et al. as a sex chromosome, due to its low autosomal coverage in males [18]. The length of chromosome 7 defined in this de novo genome assembly was also reminiscent of D. melanogaster chromosome 4. However, chromosome 7 of H. illucens was previously defined as a non-redundant sex chromosome, while chromosome 4 of D. melanogaster, also known as dot chromosome, is an ancestral X chromosome, then reverted to an autosomal chromosome, as showed by Vicoso & Bachtrog [51].

Reviewer 2 Report

Comments and Suggestions for Authors

The manuscript by Costagli et al. “De novo genome assembly at chromosome-scale of Hermetia illucens (Diptera Stratiomyidae) via PacBio and Omni-C proximity ligation technology” reports a new assembly of the H. illucens genome using PacBio reads, and Illumina reads from Omni-C techniques. In addition, authors analyze two proteins from H. illuncens that are homologues to Lsp-2 in D. melanogaster.

Comments to the authors:

Although this new genome assembly of H. illucens is of high quality, there is a previously obtained that seems to be better. I understand that one important point of this work was to test if the Omni-C techniques would be a better choice to Hi-C methods. Nevertheless, the number and type of libraries used to obtain both genomes were different, and this makes comparison difficult.

I miss an explanation of why it was chosen to analyze the structure of the Lsp-2 protein in depth. Now they seem to me to be two quite independent works. According to the title of the article, the study of Lsp-2 was not necessary, but if it has been done in such detail it should be able to contribute something important to the knowledge of the assembly or vice versa that the new assembly allowed to study these proteins with this detail. I miss the joining of the two works in the introduction. 

The paragraph on lines 258-266 led me to some confusion.

The initial draft assembly, the primary filtered assembly and the final assembly covered 99.2%, 88.9% and 89.1% of the Insecta BUSCO (Benchmarking Universal Single-Copy Orthologs v5.2.2) core genes respectively, and 95.6%, 84.6% and 84.7% of the Diptera BUSCO core genes, respectively (Figure 1). Among the core genes of Insecta covered by the final assembly 87.9% of genes were single copy (S), 1.2% of genes were duplicated (D), 1.1% of genes were fragmented (F) and 9.8% of genes were missing (M) (Figure 1); among the core genes of Diptera covered by the final assembly 83.7% of genes were single copy (S), 1.0% of genes were duplicated (D), 1.6% of genes were fragmented (F) and 13.7% of genes were missing (M) (Figure 1).

For example, the final assembly covered 89.1% of Insecta BUSCO. Then when the text says: “Among the core genes of Insecta covered by the final assembly…” I understood it to mean the 89.1 % indicated earlier. Nevertheless, there appear 9,8% of missing genes, that where not covered… In fact, the previously stated as covered refers to the sum of single copy and duplicated (87.9 + 1.2 = 89.1).

Lines 318-325. I believe this data is exactly the same as discussed in L258-266.

The soft-masked final assembly covered 89.1% and 84.7% of the Insecta and Diptera BUSCO core genes, respectively. Among the core genes of Insecta covered by the soft-masked final assembly 87.9% of genes were single copy (S), 1.2% of genes were duplicated (D), 1.1% of genes were fragmented (F) and 9.8% of genes were missing (M); among the core genes of Diptera covered by the soft-masked final assembly 83.7% of genes were single copy (S), 1.0% of genes were duplicated (D), 1.6% of genes were fragmented (F) and 13.7% of genes were missing (M).

And I have the same problem with “covered”.

I think that part of the text in the legend of figures 1 (lines 281-287), 2 (335-337) and 3 (365-366) corresponds to the discussion and should go in the main text.

What did you do with the 9.8% of sequence with no significant taxonomic identification? Were they considered contaminants or retained in the assembly? It is possible that some of these sequences are present in the tested groups but have not yet been detected or may even be novelties to the H. illucens genome.

In the discussion (lines 526-531) authors say: “The gene prediction performed by Dovetail Genomics on the soft-masked final assembly using the MAKER pipeline identified 32,516 genes. This number of predicted genes was higher than that reported in the reference genome (22,369). This was probably because each predicted gene was associated with a single transcript. In this way, individual genes coding for several alternative transcripts may have been predicted as separate genes.”

This is a great handicap and prevents doing any comparison between genomes.

L. 557-561 “Molecular analyses of hiLsp-2 may in the future shed light on the biological role of these two gene copies. ANN16409 and ANN16413 localized on chromosome 3, as the hiLsp-2 genes reported in the reference genome (LOC119651526 and LOC119651216) and as the dmeLsp-2 gene. ANN16409ANN16413, the hiLsp-2 genes and the dmeLsp-2 gene were all single-exon genes.

I am not sure what authors want to say. I can understand the importance that both copies of hiLsp-2 are in the same chromosome and in the same reported in the reference genome, but it is not clear why it is important that dmeLsp-2 is also on chromosome 3. This would be interesting if the synteny of these chromosomes is maintained between the species, but I have not seen anywhere in the manuscript that the homologies between the chromosomes of these species are discussed. In Line 253 it is said that H. illucens has seven chromosomes and I know that D. melanogaster has four chromosomes, organized in six chromosomal arms. On the other hand, maybe the period (.) after “as the dmeLsp-2 gene” should be a coma (,) and refers to the similar gene structure in both species. 

Minor comments and typos:

I have found that some abbreviations should be introduced the first time they are used to facilitate understanding by readers not introduced to the field. For example: “AED score”.

In general, avoid starting sentences with a number. There are several sentences of this kind starting on lines 246, 270, 303. For example, you can rewrite the sentence on line 303 as: “A total of 65, 65, 7671...” or “Sixty-five, 65, 7671…” I prefer the first option.

Also, in general, change the single digit numbers to letters:

L. 253. Change “7 scaffolds,” to “seven scaffolds”.

L. 301. Change “identified 7 chromosomes” to “identified seven chromosomes”.

L. 370. Change “pipeline, 2” to “pipeline, two”.

L. 488. Change “recognized 7 chromosomes” to “recognized seven chromosomes”.

L. 433. Add a comma after “as expected”.

L. 452. And 585. Change “H. illucens” to “H. illucens”.

L. 615-616. The authors thank for a photo that I have not seen anywhere.

Author Response

Comments to the authors:

Although this new genome assembly of H. illucens is of high quality, there is a previously obtained that seems to be better. I understand that one important point of this work was to test if the Omni-C techniques would be a better choice to Hi-C methods. Nevertheless, the number and type of libraries used to obtain both genomes were different, and this makes comparison difficult.

I miss an explanation of why it was chosen to analyze the structure of the Lsp-2 protein in depth. Now they seem to me to be two quite independent works. According to the title of the article, the study of Lsp-2 was not necessary, but if it has been done in such detail it should be able to contribute something important to the knowledge of the assembly or vice versa that the new assembly allowed to study these proteins with this detail. I miss the joining of the two works in the introduction. 

R: We thank the reviewer for drawing attention to the lack of connection between the work regarding the de novo genome assembly and the focus on Lsp-2 gene annotation and characterization of the Lsp-2 protein structure. To better clarify the rationale behind choosing Lsp-2 gene for annotation and protein characterization, we added some information in the introduction and in the “Lsp-2” paragraph of the discussion. For the benefit of the reviewers, the text added in the revised manuscript are also reported here.

Introduction

As described above, the use of H. illucens as animal feed is very promising due to its excellent nutritional properties for various livestock diets. However, for this insect species, prospects of genome editing aimed at the insertion of exogenous gene sequences are highly interesting, either to further improve its nutritional characteristics or to make it a vehicle for other proteins not normally expressed by H. illucens but associated with positive and beneficial, when not even protective or anti-inflammatory, properties [20]. To ensure high-level expression in tissues and life cycle stages of greater interest for animal feed use, while minimizing the impact that the expression of an exogenous protein could have on the insect's normal physiology, it is desirable that these genome editing strategies be site-specific to ensure extremely controlled temporal expression. Specifically, considering that the life cycle stages of H. illucens most frequently proposed for use as animal feed are the prepupa and mature larva, it would be important to ensure expression of the exogenous genes in these life cycle stages through site-specific genome editing at gene sequences characterized by this expression profile. Searching for genes with this expression profile in D. melanogaster on the FlyBase database (http://www.flybase.org, 10.1093/genetics/iyac035), among the genes of interest, Larval serum protein 2 (dmeLsp-2, Gene ID: 45326) was selected. DmeLsp-2 has a gene expression profile coherent with that described above, and it encodes a single protein isoform, which can be an advantage, considering that genes coding for several isoforms can more easily be associated with unknown expression regulatory systems that can cause different expression from the desired one. For this reason, in the H. illucens de novo genome assembly reported in this work, those genes with significant similarity to dmeLsp-2 were more thoroughly investigated.

Discussion

Lsp-2

Considering future genetic engineering applications aimed at improving the nutritional properties of H. illucens, the identification and characterization of endogenous genes to be used as targets for future site-specific genome editing strategies is of particular interest. Due to its observed temporal expression profile in D. melanogaster, the Lsp-2 gene emerged as an interesting target among such candidate genes [55].  

The paragraph on lines 258-266 led me to some confusion.

The initial draft assembly, the primary filtered assembly and the final assembly covered 99.2%, 88.9% and 89.1% of the Insecta BUSCO (Benchmarking Universal Single-Copy Orthologs v5.2.2) core genes respectively, and 95.6%, 84.6% and 84.7% of the Diptera BUSCO core genes, respectively (Figure 1). Among the core genes of Insecta covered by the final assembly 87.9% of genes were single copy (S), 1.2% of genes were duplicated (D), 1.1% of genes were fragmented (F) and 9.8% of genes were missing (M) (Figure 1); among the core genes of Diptera covered by the final assembly 83.7% of genes were single copy (S), 1.0% of genes were duplicated (D), 1.6% of genes were fragmented (F) and 13.7% of genes were missing (M) (Figure 1).

For example, the final assembly covered 89.1% of Insecta BUSCO. Then when the text says: “Among the core genes of Insecta covered by the final assembly…” I understood it to mean the 89.1 % indicated earlier. Nevertheless, there appear 9,8% of missing genes, that where not covered… In fact, the previously stated as covered refers to the sum of single copy and duplicated (87.9 + 1.2 = 89.1).

Lines 318-325. I believe this data is exactly the same as discussed in L258-266.

The soft-masked final assembly covered 89.1% and 84.7% of the Insecta and Diptera BUSCO core genes, respectively. Among the core genes of Insecta covered by the soft-masked final assembly 87.9% of genes were single copy (S), 1.2% of genes were duplicated (D), 1.1% of genes were fragmented (F) and 9.8% of genes were missing (M); among the core genes of Diptera covered by the soft-masked final assembly 83.7% of genes were single copy (S), 1.0% of genes were duplicated (D), 1.6% of genes were fragmented (F) and 13.7% of genes were missing (M).

And I have the same problem with “covered”.

R: We apologize for the confusion in both paragraphs and we thank the reviewer for highlighting these points, thus providing us the chance to clarify and improve the manuscript. When we state that the final assembly covered 89.1% of Insecta BUSCO, we consider the genes that have been identified as complete (thus the sum of single copy and duplicated). In the following period, we further defined the results of the BUSCO analysis, specifying, concerning the entirety of genes that are part of the BUSCO dataset, how many are complete single copy, how many are complete duplicates, how many are fragmented, and how many are missing. To better clarify the text, we rewrote such a text and removed the word 'covered' in the following chapters of the results section: “Genome library construction and sequencing and genome assembly” and “Genome Annotation”.

I think that part of the text in the legend of figures 1 (lines 281-287), 2 (335-337) and 3 (365-366) corresponds to the discussion and should go in the main text.

R: We thank the reviewer 2 for this comment. What is described in the legend of Figures 1, 4 and 5 was indeed already mentioned in the discussion. Therefore, we have removed these parts from the figure captions.

What did you do with the 9.8% of sequence with no significant taxonomic identification? Were they considered contaminants or retained in the assembly? It is possible that some of these sequences are present in the tested groups but have not yet been detected or may even be novelties to the H. illucens genome.

R: As part of the pipeline defined by Dovetail Genomics, the 9.8% of sequences with no significant taxonomic identification were considered contaminants and thus removed from the assembly. Since they are not associated with arthropod taxonomic identification, it seems unlikely to us that these sequences were inherent to H. illucens and were novelties of this de novo genome assembly. Thus, these sequences were removed for the following steps of genome assembly for this reason. To further clarify this aspect, we have specified it in the text as well.

In the discussion (lines 526-531) authors say: “The gene prediction performed by Dovetail Genomics on the soft-masked final assembly using the MAKER pipeline identified 32,516 genes. This number of predicted genes was higher than that reported in the reference genome (22,369). This was probably because each predicted gene was associated with a single transcript. In this way, individual genes coding for several alternative transcripts may have been predicted as separate genes.”

This is a great handicap and prevents doing any comparison between genomes.

R: We agree with the points raised by the referee. The initial identification of the 32,516 genes using the MAKER pipeline still represents a preliminary step in genomic annotation, which will need to be further developed. We have thus specified in the text how this represents a limitation of our gene annotation, which, despite being an important initial step, will need to be further optimized.

  1. 557-561 “Molecular analyses of hiLsp-2may in the future shed light on the biological role of these two gene copies. ANN16409and ANN16413 localized on chromosome 3, as the hiLsp-2 genes reported in the reference genome (LOC119651526 and LOC119651216) and as the dmeLsp-2 gene. ANN16409ANN16413, the hiLsp-2 genes and the dmeLsp-2 gene were all single-exon genes.”

I am not sure what authors want to say. I can understand the importance that both copies of hiLsp-2 are in the same chromosome and in the same reported in the reference genome, but it is not clear why it is important that dmeLsp-2 is also on chromosome 3. This would be interesting if the synteny of these chromosomes is maintained between the species, but I have not seen anywhere in the manuscript that the homologies between the chromosomes of these species are discussed. In Line 253 it is said that H. illucens has seven chromosomes and I know that D. melanogaster has four chromosomes, organized in six chromosomal arms. On the other hand, maybe the period (.) after “as the dmeLsp-2 gene” should be a coma (,) and refers to the similar gene structure in both species. 

R: We thank the reviewer for pointing out an unclear message in our text. The localization of dmeLsp-2 gene into chromosome 3 of D. melanogaster does not have any relevant significance as compared to the localization of hiLsp-2 genes into the H. illucens chromosome 3. On the other hand, the important point that we wanted to stress in our manuscript is that both the hiLsp-2 genes are located on chromosome 3. In order to avoid confusion, we removed “and as the dmeLsp-2 gene” from the discussion and we modified the following statement.

Minor comments and typos:

I have found that some abbreviations should be introduced the first time they are used to facilitate understanding by readers not introduced to the field. For example: “AED score”.

R: We agree on the need to specify what the AED abbreviation signifies initially, and we thank the reviewer 2 for this note. We have corrected this and other abbreviations accordingly.

In general, avoid starting sentences with a number. There are several sentences of this kind starting on lines 246, 270, 303. For example, you can rewrite the sentence on line 303 as: “A total of 65, 65, 7671...” or “Sixty-five, 65, 7671…” I prefer the first option.

R. We thank the reviewer for the suggestion and we corrected it accordingly.

Also, in general, change the single digit numbers to letters:

253. Change “7 scaffolds,” to “seven scaffolds”.

301. Change “identified 7 chromosomes” to “identified seven chromosomes”.

370. Change “pipeline, 2” to “pipeline, two”.

488. Change “recognized 7 chromosomes” to “recognized seven chromosomes”.

R: We have made the changes as suggested

433. Add a comma after “as expected”.

We have made the change as suggested

452 and 585. Change “H. illucens” to “H. illucens”.

We have made the change as suggested

615-616. The authors thank for a photo that I have not seen anywhere.

The picture in the submission was in the graphical abstract. Now, as also requested by the Reviewer 1, we added a figure in the manuscript text with the pictures of the instars (figure 1).